# Interactive Remote Sensing Retrieval via Dialogue-Guided Intent Refinement and Attribute Reasoning

## Abstract

Remote sensing image-text retrieval (RSITR) addresses the bidirectional retrieval problem between images and text in large-scale remote sensing databases. Despite significant progress, due to the pronounced inter-class similarity and multi-scale characteristics inherent in remote sensing (RS) imagery, current research relying on raw descriptive text often suffers from ambiguous semantics and vague user intent, thereby limiting the generalizability in real-world scenarios. To overcome these challenges, this work leverages the power of multimodal large language models (MLLMs) and creates a novel dialogue-driven cross-modal retrieval framework (DiaRet) for RSITR. DiaRet initiates a user-given query and induces multi-level semantic concepts to construct a comprehensive and deterministic understanding of the scene. In line with this promise, our method engages in a context-aware question-answer interaction to progressively clarify the vague intention. Furthermore, we introduce an LLM-based fine-grained attribute reasoning module that distills the dialogue into a structured formalism of atomic editing instructions and critical visual keywords, which enables targeted optimization and sharpens the focus on discriminative visual details. Extensive experiments on the RSICD and RSITMD benchmarks demonstrate that our DRS framework achieves state-of-the-art performance, validating the superiority of our interactive, dynamic dialogue approach for accurate RSITR.

## 1 Introduction

The rapid advancement of Earth observation technologies has made remote sensing (RS) imagery a critical resource for diverse applications such as urban planning, environmental monitoring, and disaster response (Wang et al., 2025; Li et al., 2020; 2024; Weiss et al., 2020). As the volume of RS images grows exponentially, there is an increasing demand for intelligent systems capable of retrieving relevant images from large-scale archives using natural language queries. This has led to the emergence of remote sensing image-text retrieval (RSITR) (Lu et al., 2017; Yuan et al., 2021a;b; Pan et al., 2023a; Yuan et al., 2022; Yang et al., 2024b; Sun et al., 2024; Chen et al., 2025), a cross-modal matching task that bridges the semantic gap between visual content and textual captions.

Unlike natural scenes, RS images present quite distinct characteristics, as shown in Figure 1. They often exhibit high intra-class variance (*e.g.*, populated areas can vary substantially in building density or surrounding greenery) and low inter-class variance (*e.g.*, residential and industrial zones may appear similarly as dense building clusters from an overhead view). Furthermore, RS images with numerous objects show properties at diverse scales and much background content unrelated to the caption subject. Some methods (Yuan et al., 2021a; 2022; Chen et al., 2025) address these challenges by investigating multiscale feature representations, either from regional image components or textual hierarchies, to enhance cross-modal alignment. Motivated by the success of pre-trained vision-language models (PVLM) (Radford et al., 2021; Li et al., 2022) in various downstream tasks as well as natural image-text matching, many methods (Zhang et al., 2023; Ji et al., 2023; Liu et al., 2024; Zhang et al., 2024) exploit the external knowledge from PVLMs that project both modalities into a joint embedded subspace, thus can effectively alleviate the semantic ambiguity.

All the aforementioned methods are constrained by **a single-turn static interaction paradigm**, which relies on the strong assumption that a single-shot query can precisely capture user intent or that image captions are exhaustively descriptive. In practice, user-provided descriptions are typically

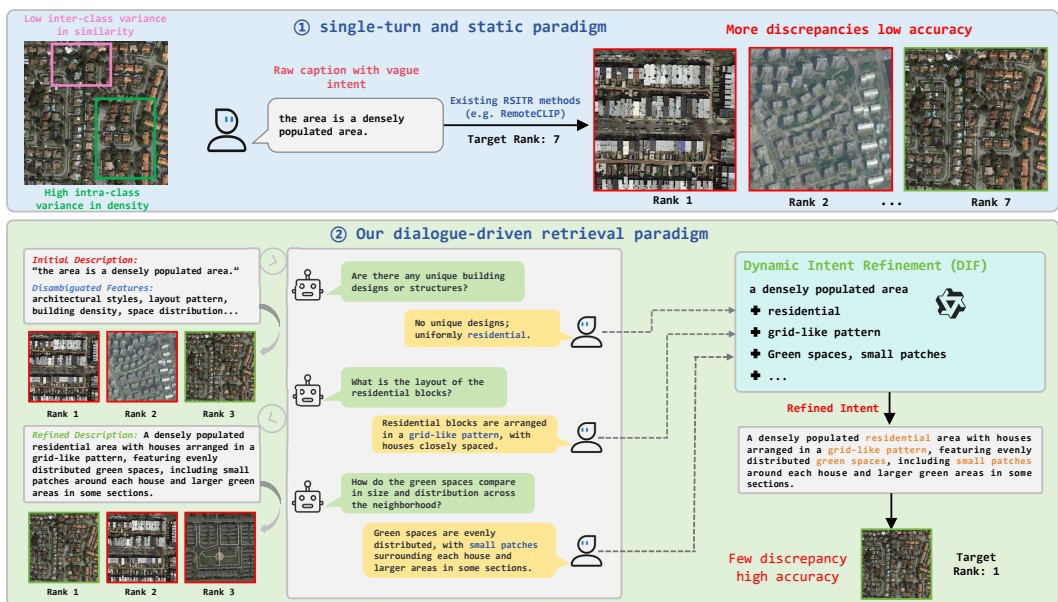

Figure 1: Motivation of DiaRet. The initial description (*e.g.*, "a densely populated area") is vague and fails to reveal fine-grained distinctions, leading to inaccurate retrieval. (1) Existing RSITR methods under a single-turn static paradigm often yield low accuracy due to vague queries. (2) Our dialogue-driven retrieval paradigm progressively clarifies user intent through context-aware questioning and refinement, uncovering discriminative attributes to achieve superior retrieval accuracy.

variable, subjective, and biased towards high-level summarization, such as "a densely populated area" or "a harbor with some trees". Such summaries lack discriminative specificity to disambiguate visually analogous scenes, leading to unreliable retrieval results.

Recent advances in multimodal large language models (MLLMs) (Li et al., 2023; Liu et al., 2023; Wang et al., 2024a), renowned for their powerful visual contextual reasoning and dense caption generation capabilities, open a promising avenue for interactive query refinement through natural language dialogue. This potential motivates the exploration of **a dialogue-driven retrieval paradigm**, where multi-turn interaction progressively uncovers discriminative, fine-grained attributes to disambiguate visually similar yet semantically distinct RS scenes. However, due to the spatial-scale variations and dense distributions of small objects in RS images, directly applying existing interactive frameworks designed for natural images (Levy et al., 2023; Lee et al., 2024; Luo et al., 2025) is challenged to capture the unique semantic and structural nuances critical for accurate alignment.

Based on the above analysis, in this work, we propose a novel dialogue-driven retrieval framework for RSITR (DiaRet). Beginning with an initial, vague textual, DiaRet presents a dynamic intent refinement (DIR) module that engages the user in a context-aware, vision-grounded question-answer interaction. Before feeding the raw description into the LLM system as existing interactive methods (Levy et al., 2023; Bai et al., 2025), we prompt an MLLM to generate hierarchical scene descriptions at different levels that explicitly model the scene context at multiple granularities. In each dialogue turn, DIR uncovers missing or implicit attributes while injecting new discriminative cues into the query representation to clarify the user intent and finally provide a more deterministic query to boost retrieval accuracy. To translate this unstructured dialogue into targeted model supervision, we introduce an LLM-based fine-grained attribute reasoning (FAR) module. It distills the dialogue history into structured and interpretable signals, which can effectively enhance the model sensitivity to fine-grained visual attributes. The main contributions of this work are as follows.

- We propose DiaRet, to our best knowledge, it is a distinct dialogue-driven retrieval framework that pioneers a multi-turn interactive paradigm for RSITR. Extensive experiments demonstrate that DiaRet consistently outperforms state-of-the-art methods.

- We propose the DIR, which uncovers missing or implicit attributes while injecting new discriminative cues into the query representation, enabling a context-aware, vision-grounded question-answer interaction to build a deterministic query of user intent.

- We propose the fine-grained attribute reasoning (FAR) module that converts this dialogue history into structured supervision signals, enabling targeted optimization of fine-grained attribute alignment through specialized loss functions.

## 2 RELATED WORK

### 2.1 REMOTE SENSING IMAGE-TEXT RETRIEVAL

RSITR aims to establish bidirectional correspondences between remote sensing images and their natural language descriptions. Early research in this domain (Abdullah et al., 2020; Lv et al., 2021; Yuan et al., 2021a) mainly relied on convolutional networks for visual feature extraction and recurrent architectures (*e.g.*, long-short term memory (LSTM) and gated recurrent unit (GRU)) for text encoding. (Abdullah et al., 2020) pioneers the task with an average fusion strategy using visual self-attention to achieve fine-grained alignment. (Yuan et al., 2021a) contributes the RSITMD dataset, which features more challenging scenes and fine-grained captions, dramatically advancing this field. KAMCL (Ji et al., 2023) proposes a knowledge-aided learning framework to reinforce key concepts and distinguish analogous descriptions. Latter methods (Wang et al., 2022; Zhang et al., 2022; Yuan et al., 2023; Chen et al., 2024) absorb the merits of transformers (Vaswani et al., 2017; Devlin et al., 2019; Dosovitskiy et al., 2020) in modeling long-range dependencies, which apply transformer encoders for both text and vision, showing promising performance in cross-modal retrieval as well as RSITR. Recently, the advent of PVLMs such as CLIP (Radford et al., 2021) and BLIP (Li et al., 2022) has revolutionized RSITR, where methods (Zhang et al., 2024; Kuckreja et al., 2024) fine-tune PVLMs in large-scale RS datasets to produce domain-specific knowledge for downstream tasks. (Yuan et al., 2023) introduces a parameter-efficient adapter into CLIP to explore the specific knowledge of RS image-text pairs. (Zhang et al., 2024) constructs the RS5M dataset with 5 million annotated pairs and presents GeoRSCLIP fine-tuned on RS5M, achieving 3%-6% improvements in RSITR. In contrast to these methods rely on large-scale pre-training, our work applies a distinct dialogue-driven framework that enables fine-grained cross-modal alignment in a data-efficient manner, offering a more practical and user-centric solution for RSITR.

### 2.2 INTERACTIVE CROSS-MODAL RETRIEVAL

The recent emergence of LLMs (Dubey et al., 2024; Wang et al., 2024a) and MLLMs (Li et al., 2023; Liu et al., 2023; Hurst et al., 2024) has demonstrated exceptional capabilities in language understanding, generation, and complex reasoning. In vision-language tasks, MLLMs are increasingly leveraged to generate rich, descriptive image captions, augmenting datasets with valuable supervisory signals beyond manual annotations. Their proficiency in contextual instruction-following has catalyzed a paradigm shift toward interactive systems, where MLLMs act as intelligent agents within dialogue-based interfaces to clarify user intent and dynamically refine model predictions (Lee et al., 2024; Qin et al., 2025; Bai et al., 2025). By engaging users in a conversational loop, these methods leverage dialogue history to pose context-aware questions, dynamically refining the search query with each turn. For instance, ChatIR (Levy et al., 2023) employs LLMs to simulate dialogues between users and answer bots, compensating for the scarcity of specialized text-to-image datasets tailored. PlugIR (Lee et al., 2024) reformulates the LLM-based dialogue context and devises a questioner to remove redundant questions in such a dialogue system. ICL (Qin et al., 2025) proposes an interactive cross-modal learning framework to solve the text-to-image person re-identification problem through human-centered interaction at test time. Differently, this work targets a more fine-grained task of RSITR, where retrieving relevant images from large galleries requires highly discriminative queries. Besides, we introduce the DIR and FAR that convert natural language dialogue into structured instructions, facilitating the learning of discriminative RS visual features.

## 3 METHODOLOGY

Given an image-text pair $\{(I_i, T_i)\}$, where $I_i$ denotes the RS image and $T_i$ is its initial caption, we propose DiaRet, which progressively uncovers user intent through visual-grounded dialogue. As shown in Figure 2, DiaRet is composed of two tightly coupled modules: 1) Dynamic intent refinement (DIR) generates context-aware questions and image-grounded answers to clarify user

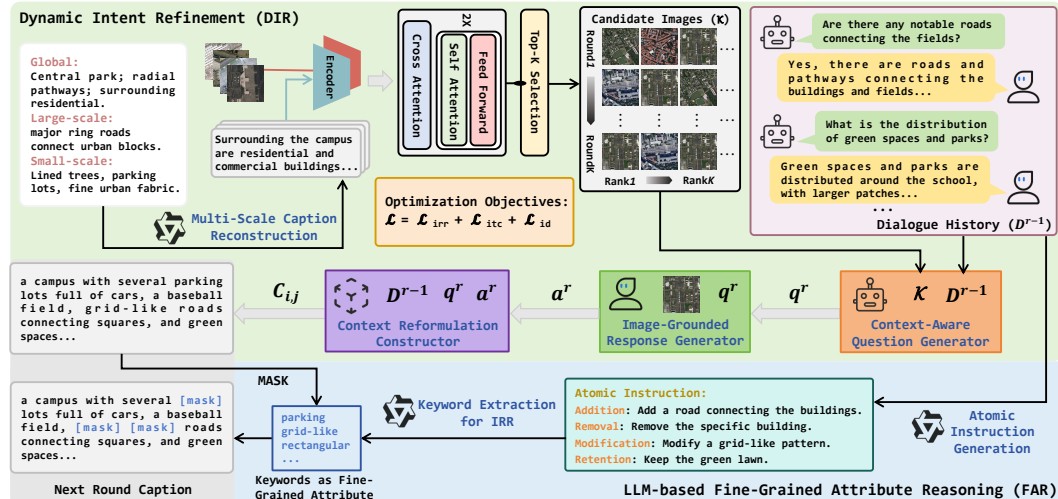

Figure 2: Illustration of the proposed DiaRet Framework, which employs an interactive dialogue paradigm where context-aware questioning progressively refines user intent, leading to precise retrieval results. Key steps include candidate image selection, reformulation of captions based on dialogue interactions, and generation of atomic instructions for fine-grained alignment.

intent iteratively; and 2) Fine-grained attribute reasoning (FAR) transforms dialogue history into structured instructions and extracts critical visual keywords for supervised alignment.

## 3.1 DYNAMIC INTENT REFINEMENT

The goal of DIR aims at generating discriminative questions and image-grounded responses to refine ambiguous queries through multi-turn vision–language interactions. The initial caption $T_i$ is regarded as the first-round textual input. DiaRet proceeds through $N$ rounds of dialogue, generating a caption $C_{i,j}$ at each round $j$ progressively enriched through dialogue, and encoded into global and local features for retrieval.

**Multi-Scale Caption Reconstruction:** As illustrated in Figure 3, RS images typically contain large-scale variations (*e.g.*, urban regions, airports, highways) as well as dense distributions of small objects (*e.g.*, vehicles, trees). A generally vague caption is often insufficient to capture the complex semantics. Therefore, instead of directly feeding the raw de-

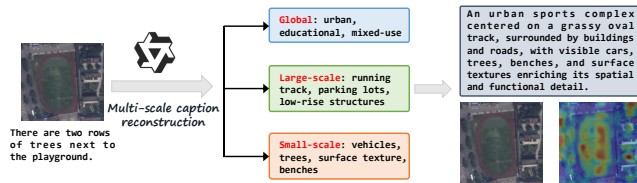

Figure 3: Illustration of multi-scale caption reconstruction.

scription into the LLM system as existing interactive methods (Levy et al., 2023; Bai et al., 2025), we propose to induce multi-scale concepts into the raw captions. Specifically, we prompt an MLLM $\mathcal{M}_{\text{multi-scale}}$ to generate hierarchical scene descriptions at three levels: (i) *global scene semantics* (*e.g.*, "urban, educational, mixed-use"), (ii) *large-scale structures* (*e.g.*, "ring roads, playgrounds, parking lots"), and (iii) *small-scale details* (*e.g.*, "vehicles, trees, benches, surface textures"). These hierarchical descriptions are concatenated with the initial caption $T_i$ to form $c_1$, which serves as a comprehensive starting point for subsequent questioning. By explicitly modeling the scene at multiple granularities, this module enriches the query representation, enhances interpretability, and provides stronger semantic grounding for dynamic intent completion.

**Context-Aware Question Generator.** To reduce semantic ambiguity caused by the geographical and diverse semantics of RS scenes, the context-aware question generator creates discriminative questions focused on high-level, visually distinguishable features.

At dialogue round $r$, we first retrieve the top-$S$ most relevant images based on the current caption embedding $c_{r-1}$:

$$\mathcal{S} = \text{topS}\left(\text{sim}(c_{r-1}, v_i)\right), \tag{1}$$

where $\text{sim}(\cdot, \cdot)$ denotes cosine similarity. We apply multi-head cross-attention using $c_{r-1}$ as the query and candidate image features as key-value pairs. A Transformer encoder refines the attended features:

$$w = \text{Transformer}(s_i; c_{r-1}), \quad \mathcal{K} = \text{topK}(\text{softmax}(w)), \tag{2}$$

where $\mathcal{K}$ denotes the final informative candidates. Then, an MLLM $\mathcal{M}_{\text{question}}$, guided by a prompt $\mathcal{T}_{\text{question}}$, uses $\mathcal{K}$ and dialogue history $D^{r-1}$ to generate a novel question $\hat{q}^r$:

$$\hat{q}^r = \mathcal{M}_{\text{question}}\left(\mathcal{T}_{\text{question}}(\mathcal{K}, D^{r-1})\right). \tag{3}$$

**Image-Grounded Response Generator**. Given $\hat{q}^r$, a simulated response $\hat{a}^r$ is produced:

$$\hat{a}^r = \mathcal{M}_{\text{response}}(\mathcal{T}_{\text{response}}(I_i, \hat{q}^r)), \tag{4}$$

where $\mathcal{T}_{\text{response}}$ encodes the visual context. The response strictly adheres to the visible content in the RS image, using domain-specific terminology while avoiding unverifiable inferences.

**Context Reformulation Constructor**. Each $(\hat{q}^r, \hat{a}^r)$ pair is appended to history $D^r$ and reformulated into a concise caption $C_{i,j}$, which is re-encoded for the next round.

In summary, through such an iterative refinement, DIR converts vague queries into precise, visually grounded descriptions. It not only identifies ambiguous cues by attending to informative candidates, but also supports structured QA interactions instead of ad-hoc rewriting.

### 3.2 LLM-based Fine-Grained Attribute Reasoning

User-provided RS queries are often coarse, omitting spatial, structural, or material details. While the DIR module enriches them through dialogue, the outputs remain unstructured and sometimes redundant. We thereby introduce the FAR module, which distills dialogue history into explicit, machine-interpretable cues and highlights discriminative attributes for supervision. These cues serve as privileged inputs to the IRR loss (Jiang & Ye, 2023), ensuring training focuses on contextually relevant features. The FAR module operates in two sequential stages: Atomic Instruction Generation and Keyword Extraction for implicit relation reasoning (IRR).

**Atomic Instruction Generation**. The dialogue responses are parsed by an LLM into atomic editing instructions $O_{i,j} = \{o_{i,j}^1, \ldots, o_{i,j}^M\}$, categorized as:

- *Addition*: Add visible elements (*e.g.*, "Add visible pathways around the baseball field").
- *Removal*: Exclude distractors (*e.g.*, "Remove white houses beside the field").
- *Modification*: Adjust visual attributes (*e.g.*, "Change the fence to a wooden barrier").
- *Retention*: Emphasize key identity cues (*e.g.*, "Keep the green lawn").

**Keyword Extraction for IRR.** Each instruction $o_{i,j}^m$ is reformulated into an attribution sentence $A_{i,j}$ (*e.g.*, "There are visible pathways around the baseball field"). From these, FAR extracts salient visual keywords (*e.g.*, "pathways", "wooden fence") while filtering out uninformative tokens. These keywords guide the IRR loss (Jiang & Ye, 2023) to mask and predict fine-grained, visually grounded terms, driving cross-modal alignment on discriminative attributes and reducing query ambiguity.

### 3.3 Optimization Objectives

Our model is optimized with three complementary objectives: (i) the IRR loss $\mathcal{L}_{\text{irr}}$ (Jiang & Ye, 2023) that applies masked language modeling on dialogue-derived keywords to promote fine-grained alignment, (ii) the *image-text contrastive* (ITC) loss $\mathcal{L}_{\text{itc}}$ that maximizes similarity between matched image–text pairs while minimizing mismatches, and (iii) the *identity* (ID) loss $\mathcal{L}_{\text{id}}$ (Zheng et al., 2020) that enforces intra-modal consistency by grouping embeddings of the same scene.

For the IRR loss, given a caption $C_{i,j}$ and corresponding attribution sentences $A_{i,j}$, the masked tokens are fused with local visual features $V_i^{local}$ through a multimodal encoder. The encoder produces contextual embeddings $\{h_i\}_{i=1}^{L}$, which are used to predict the masked tokens:

$$\mathcal{L}_{\text{irr}} = \frac{1}{|\mathcal{M}||\mathcal{V}|} \sum_{i \in \mathcal{M}} \sum_{j \in |\mathcal{V}|} y_{i,j} \log \frac{\exp(m_{i,j})}{\sum_{k=1}^{|\mathcal{V}|} \exp(m_{i,k})}, \tag{5}$$

where $\mathcal{M}$ is the set of masked keywords, $\mathcal{V}$ is the vocabulary, $m_{i,j}$ is the predicted score, and $y_{i,j}$ is the ground truth. The ITC loss separates matched and unmatched pairs in a batch of size $B$. For each image $I_i$ and caption $C_{i,j}$, it maximizes cosine similarity for positives while minimizing it for negatives:

$$S_{\text{i2t}}(I_i) = \frac{\exp(\text{sim}(v_i, c_{i,j})/\tau)}{\sum_{k=1}^{B} \exp(\text{sim}(v_i, c_{k,j})/\tau)}, \tag{6}$$

$$S_{\text{t2i}}(C_{i,j}) = \frac{\exp(\text{sim}(c_{i,j}, v_i)/\tau)}{\sum_{k=1}^{B} \exp(\text{sim}(c_{i,j}, v_k)/\tau)}, \tag{7}$$

$$\mathcal{L}_{\text{itc}} = -\frac{1}{B} \sum_{i=1}^{B} \left[ \log S_{\text{i2t}}(I_i) + \log S_{\text{t2i}}(C_{i,j}) \right], \tag{8}$$

where $\tau$ is a learnable temperature parameter. The ID loss (Zheng et al., 2020) enforces intra-modal consistency by grouping images and captions from the same RS scene identity. A softmax classifier encourages embeddings from the same identity to cluster together:

$$\mathcal{L}_{\text{id}} = -\frac{1}{B} \sum_{i=1}^{B} \log \frac{\exp(W_{y_i} \cdot v_i^{cls})}{\sum_{k=1}^{K} \exp(W_k \cdot v_i^{cls})}, \tag{9}$$

Overall, the full training objective combines the above three as

$$\mathcal{L} = \mathcal{L}_{\text{irr}} + \mathcal{L}_{\text{itc}} + \mathcal{L}_{\text{id}}. \tag{10}$$

## 4 EXPERIMENTS

### 4.1 EXPERIMENT SETUP

**Datasets.** We evaluate our DiaRet using two benchmark datasets: RSICD (Lu et al., 2017) and RSITMD (Yuan et al., 2021a). The RSICD dataset contains 10,921 images, each with dimensions of 224×224 pixels, paired with textual descriptions. The RSITMD dataset comprises 4,743 images, each measuring 256×256 pixels, also paired with textual captions. Following the protocol in (Yuan et al., 2021a), we split both datasets into training (80%), validation (10%), and test (10%) sets.

**Evaluation Metrics.** We adopt Recall at K (R@K, K=1,5,10) and mean Recall (mR) as evaluation metrics for both image-to-text retrieval and text-to-image retrieval tasks, consistent with standard RSITR evaluation protocols. R@K measures the percentage of ground-truth instances retrieved within the top k results, providing insight into retrieval precision at different ranks. The mR metric computes the average of all six R@K scores (three for image-to-text retrieval and three for text-to-image retrieval), offering a comprehensive measure of overall performance.

**Implementation Details**. All experiments are implemented using PyTorch and conducted on a single NVIDIA GeForce RTX 4090 GPU. The vision encoder and text encoder are initialized with CLIP pre-trained weights (ViT-B/16), producing 512-dimensional embeddings for both image and text features. Input images are resized to 256×256 pixels, and the maximum query text length is set to 77 tokens. The model is optimized using the ADAM optimizer (Kinga et al., 2015) with a batch size of 64 and trained for 20 epochs. For dialogue-driven retrieval, we employ Qwen-VL-Max as the MLLM to perceive image content and generate context-aware responses during the interaction process. The textual reasoning and attribute extraction components are supported by Qwen-Max, serving as the backbone LLM for fine-grained attribute reasoning and instruction generation. We conduct 5 rounds of dialogue per query, as this setting achieves a favorable balance between retrieval performance and interaction cost.

Table 1: Comparison results of image-to-text and text-to-image retrieval on RSICD and RSITMD datasets. Best results in each group are bold.

| Methods | Backbone | RSICD | | | | | | | RSITMD | | | | | | |
| | | I2T | | | T2I | | | mR | I2T | | | T2I | | | mR |
| | | R@1 | R@5 | R@10 | R@1 | R@5 | R@10 | | R@1 | R@5 | R@10 | R@1 | R@5 | R@10 | |
| ❶ Traditional / Non-VLP-based Methods | | | | | | | | | | | | | | | |
| LW-MCR (Yuan et al., 2021b) | SqueezeNet | 3.29 | 12.52 | 19.93 | 4.66 | 17.51 | 30.02 | 14.66 | 10.18 | 28.98 | 39.82 | 7.79 | 30.18 | 49.78 | 27.79 |
| AMFMN (Yuan et al., 2021a) | ResNet50 | 5.39 | 15.08 | 23.40 | 4.90 | 18.28 | 31.44 | 16.42 | 11.06 | 29.20 | 38.72 | 9.96 | 34.03 | 52.96 | 29.32 |
| GaLR (Yuan et al., 2022) | ResNet18 | 6.59 | 19.85 | 31.04 | 4.69 | 19.48 | 32.13 | 18.96 | 14.82 | 31.64 | 42.48 | 11.15 | 36.68 | 51.68 | 31.41 |
| SWAN (Pan et al., 2023b) | ResNet50 | 7.41 | 20.13 | 30.86 | 5.56 | 22.26 | 37.41 | 20.61 | 13.35 | 32.15 | 46.90 | 11.24 | 40.40 | 60.60 | 34.11 |
| PIR (Pan et al., 2023a) | ResNet50 | 9.88 | 27.26 | 39.16 | 6.97 | 24.56 | 38.92 | 24.46 | 18.14 | 41.15 | 52.88 | 12.17 | 41.68 | 63.41 | 38.24 |
| KAMCL (Ji et al., 2023) | ResNet50 | 12.08 | 27.26 | 38.70 | 8.65 | 27.43 | 42.51 | 26.10 | 16.51 | 36.28 | 49.12 | 13.50 | 42.15 | 59.32 | 36.14 |
| MSA (Yang et al., 2024b) | ResNet50 | 10.16 | 25.71 | 36.96 | 7.87 | 25.67 | 41.85 | 24.70 | 22.35 | 42.92 | 55.75 | 15.18 | 47.35 | 64.73 | 41.38 |
| **DiaRet (Ours)★** | ResNet50 | 12.70 | 25.86 | 37.20 | 11.49 | 32.01 | 47.86 | 27.85 | 22.78 | 41.15 | 55.97 | 18.00 | 50.70 | 68.71 | 42.89 |
| ❷ VLP-based Methods | | | | | | | | | | | | | | | |
| RemoteCLIP (Liu et al., 2024) | ViT-L/14 | 18.39 | 37.42 | 51.05 | 14.73 | 39.93 | 56.58 | 36.35 | 28.76 | 52.43 | 63.94 | 23.76 | 59.51 | 74.73 | 50.52 |
| PE-RSITR (Yuan et al., 2023) | ViT-B/32 | 14.13 | 31.51 | 44.78 | 11.63 | 33.92 | 50.73 | 31.12 | 23.67 | 44.07 | 60.36 | 20.10 | 50.63 | 67.97 | 44.47 |
| AIR (Yang et al., 2024c) | ViT-B/16 | 18.85 | 39.07 | 51.78 | 14.24 | 39.03 | 54.49 | 36.24 | 29.20 | 49.78 | 65.27 | 26.06 | 57.04 | 73.98 | 50.22 |
| CUP (Wang et al., 2024b) | ViT-L/14 | 13.14 | 36.14 | 51.36 | 12.17 | 35.36 | 52.45 | 33.43 | 23.23 | 45.80 | 60.84 | 19.84 | 52.83 | 71.11 | 45.61 |
| CLGSA (Chen et al., 2025) | ViT-B/16 | 11.75 | 34.07 | 50.12 | 14.38 | 36.73 | 45.35 | 32.07 | 22.52 | 52.39 | 69.46 | 26.77 | 48.23 | 60.39 | 46.63 |
| IERR (Yang et al., 2024a) | ViT-B/16 | 17.29 | 35.41 | 48.58 | 13.07 | 35.32 | 51.31 | 33.50 | 27.66 | 48.89 | 61.06 | 24.43 | 56.59 | 72.66 | 48.55 |
| ❸ Dialogue-driven Methods | | | | | | | | | | | | | | | |
| ChatIR (Levy et al., 2023) | ViT-B/16 | 19.39 | 37.78 | 51.22 | 14.71 | 39.34 | 55.99 | 36.40 | 28.31 | 51.32 | 64.15 | 24.64 | 56.81 | 73.62 | 49.81 |
| PlugIR (Lee et al., 2024) | ViT-B/16 | 20.40 | 39.52 | 51.69 | 16.54 | 41.33 | 57.25 | 37.79 | 30.08 | 51.99 | 64.60 | 23.00 | 57.47 | 74.29 | 50.24 |
| **DiaRet (Ours)** | ViT-B/16 | **21.95** | **43.27** | **56.45** | **16.43** | **41.68** | **57.91** | **39.61** | **32.30** | **53.76** | **67.69** | **26.99** | **58.98** | **76.41** | **52.69** |

## 4.2 COMPARISONS WITH STATE-OF-THE-ART METHODS

We conduct a comprehensive comparison of our DiaRet against a wide range of state-of-the-art (SOTA) methods on the RSICD and RSITMD datasets, including LW-MCR (Yuan et al., 2021b), AMFMN (Yuan et al., 2021a), GaLR (Yuan et al., 2022), SWAN (Pan et al., 2023b), PIR (Pan et al., 2023a), KAMCL (Ji et al., 2023), MSA (Yang et al., 2024b), RemoteCLIP (Liu et al., 2024), PE-RSITR (Yuan et al., 2023), AIR (Yang et al., 2024c), CUP (Wang et al., 2024b), CLGSA (Chen et al., 2025), IERR (Yang et al., 2024a), ChatIR (Levy et al., 2023), and PlugIR (Lee et al., 2024).

As shown in Table 1, our approach establishes new SOTA on both datasets. To be specific, on the RSICD dataset, our method achieves an mR of 39.61%, significantly outperforming the previous best (PlugIR (Lee et al., 2024)) by 1.82 points. More notably, on the RSITMD dataset, we achieve an mR of 52.69%, surpassing the most SOTA method RemoteCLIP (Liu et al., 2024) by 2.17 points, even though it utilizes additional training data (Ret-3, DET-10, and SEG-4) as defined in their work. This substantial improvement is particularly evident in the higher recall thresholds (R@5 and R@10), indicating that our model is highly effective at ranking relevant results at the top of the list.

## 4.3 ANALYTIC AND ABLATION STUDY

**Effectiveness of Dialogue Rounds**. As shown in Figure 4, the mR score is low at 0–1 rounds, directly reflecting the initial query ambiguity. This confirms that a static single-turn paradigm is inadequate for the RS domain, where high intra-class variance and low inter-class variance make vague queries prone to failure. Within the first five rounds, performance rises sharply. This gain stems from the DIR module, which progressively uncovers fine-grained attributes through context-aware questioning and image-grounded responses. The mR increase thus quantifies the ability of our method for user intent refinement. After the fifth round, improvements saturate, indicating that most queries can be sufficiently refined within five interactions. Fixing the number of rounds to five achieves a balance between accuracy and cost, yielding 39.61% on RSICD and 52.69% on RSITMD. These results empirically support our central claim: shifting from static single-turn retrieval to a dialogue-based paradigm is essential to resolve query ambiguity and SOTA performance.

**Comparison with Different Intent Refinement Variants**. We further compare three variants of the DIR module, as shown in Figure 5. **1) Caption Rewriting Only.** This variant removes interaction and relies on an LLM to polish the initial caption, producing the lowest scores (I2T R@1 = 27.21%, T2I R@1 = 23.80%, mR = 49.06%). **2) Random Questions.** This variant keeps multi-turn interaction but replaces context-aware questioning with random templates. Its performance (I2T R@1 = 31.85%, T2I R@1 = 23.49%, mR = 50.78%) is slightly higher than rewriting, yet still limited. Random, ungrounded questions cannot form a coherent refinement process or ensure new discrim-

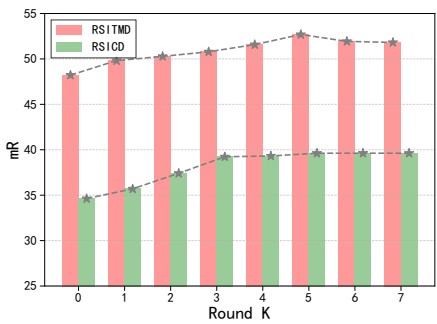

Figure 4: Performance (mR) comparison over dialogue rounds on RSITMD and RSICD datasets.

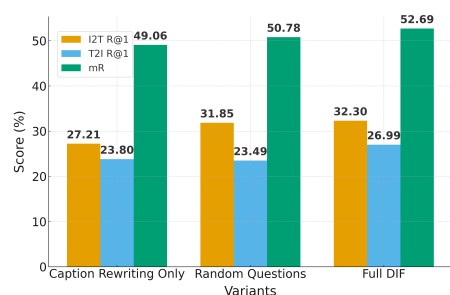

Figure 5: Performance comparison of different DIR variants on RSITMD.

Table 2: Ablation study on LLM-based Fine-Grained Attribute Reasoning on RSITMD Test Set. DK: Direct Keywords; FAR: LLM-based Fine-Grained Attribute Reasoning.

| Methods | I2T R@1 | T2I R@1 | mR |
|---|---|---|---|
| w/o IRR | 30.74 | 25.44 | 51.57 |
| IRR (DK) | 31.55 | 26.21 | 52.38 |
| IRR (FAR) | **32.30** | **26.99** | **52.69** |

Table 3: Ablation study on multiscale feature extraction on RSITMD Test Set. G: Global, L: Large-Scale, S: Small-Scale.

| No. | G | L | S | R@1 | mR |
|---|---|---|---|---|---|
| #1 | ✗ | ✗ | ✗ | 31.41 / 26.06 | 52.44 |
| #2 | ✔ | ✗ | ✗ | 29.28 / 25.88 | 52.14 |
| #3 | ✔ | ✔ | ✗ | 31.86 / 25.75 | 52.52 |
| #4 | ✔ | ✔ | ✔ | **32.30 / 26.99** | **52.69** |

inative cues each round. **3) Our Full DIR.** As we can see, the full model achieves the best results (I2T R@1 = 32.30%, T2I R@1 = 26.99%, mR = 52.69%). This success stems from its proactive, structured vision–language loop: each round identifies informative candidates and poses targeted questions, ensuring iterative discovery of missing attributes.

**Effectiveness of Multi-scale Feature Extraction in DIR**. RS imagery is characterized by large-scale variations, dense distributions of small objects, and unique spatial configurations, as emphasized in the introduction. Single-granularity descriptions cannot capture such complexity, making a multiscale feature extraction strategy necessary to jointly model global layouts and fine details. The ablation in Table 3 shows a clear progression. Without multi-scale features (#1), the baseline reaches mR 52.44%. Using only global cues (#2) reduces performance (52.14%), since coarse descriptions ignore internal structure and are sensitive to high intra-class variance. Adding large-scale features (#3) raises mR to 52.52%, confirming that modeling major scene components helps distinguish globally similar scenes. The full model further incorporates additional small-scale features (#4) achieves the best performance (52.69%), validating that hierarchical feature extraction across global, large-scale, and small-scale levels is essential to handle the multiscale nature of RS imagery to achieve robust cross-modal retrieval.

**Effectiveness of LLM-based FAR**. In our DiaRet, the FAR module serves as a translator, converting dialogue history into structured, machine-interpretable supervision. We compare three variants: **1) w/o IRR Loss:** Removing fine-grained alignment drops mR to 51.57%, showing that coarse matching alone cannot handle the high intra-class variance of remote sensing imagery. **2) IRR (DK):** Directly extracting keywords improves mR to 52.38 confirming that dialogue contains useful cues but also introducing noise from redundant or non-visual expressions. **3) IRR (FAR):** The full FAR pipeline reaches the best result (mR = 52.69%) by first structuring dialogue into atomic instructions (Addition, Removal, Modification, Retention) and then refining keywords, thus filtering noise and providing targeted supervision. These results indicate that our FAR can effectively transform raw dialogue into precise and interpretable signals.

### 4.4 QUALITATIVE RESULTS

We present qualitative results of top-5 retrievals for both text-to-image and image-to-text retrieval on the RSITMD dataset. As shown in Figure 6 and Figure 7, our model retrieves semantically consistent image-text pairs, demonstrating effective cross-modal alignment in remote sensing scenarios. In

| Query | Top-5 Images |
|-------|-------------|
| There are some coloured buildings standing on both sides of this main road. |  |
| Buildings, ponds, swimming pools and resorts are near the beach. |  |
| surrounded by some blue buildings there is a square with some rectangle patterns on the ground. |  |

Figure 6: Top-5 retrieved results for text-to-image on RSITMD Test Set.

| Query | Top-5 Captions |
|-------|----------------|
|  | The red rectangular center is built between the railway and the car road.
The red rectangular central building is located between the railway and the highway with cars.
A red rectangular central building is built between the railway and a highway with cars.
The red rectangular central building is between the railway and the street with cars.
A red rectangular public building between a railway and a car. |
|  | Two rows of tanks and several other storage tanks were on the bare ground near the road.
Two rows of storage tanks and several others were on the bare ground near the highway.
Four rows of white and black fuel tanks are arranged neatly.
Two rows of storage tanks and several other storage tanks are in a bare field near the road.
Two rows of oil tankers and several rows on the bare ground along the road. |
|  | a striped termial building sits on a large apron.
a prolate ellipse boarding gate next to the parking apron.
we can see a z shaped termial building sits on the apron which is surrounded by runways.
Several planes were parked near the three terminals.
a zigzag boarding gate ande some white blue planes. |

Figure 7: Top-5 retrieved results for image-to-text on RSITMD Test Set.

**text-to-image retrieval**, the model accurately captures the intent of diverse queries, retrieving images that reflect both global scene layouts and fine-grained spatial configurations. For **image-to-text retrieval**, the captions closely describe the visual content, capturing dominant land-use patterns, infrastructure types, and structural characteristics. In most cases, ground-truth results frequently appear in the top-1 and top-2 results, while lower-ranked entries maintain partial relevance (*e.g.*, matching scene type or dominant structure), indicating robustness to semantic variations and confirming that our method effectively bridges the gap between natural language and RS imagery (**More qualitative results are shown in Appendix**A.2).

## 5 CONCLUSION

In this paper, we propose a novel dialogue-driven retrieval framework for remote sensing, DiaRet, which transforms static image-text retrieval into an interactive process. To address the ambiguity of initial queries and the scarcity of fine-grained annotations, DiaRet leverages MLLMs to engage in multi-round dialogues with users, progressively clarifying their intent through context-aware questioning. The dialogue history is used to reconstruct refined textual descriptions and, more importantly, to drive a FAR module that enables targeted learning on discriminative visual features. Additionally, a multi-scale feature learning strategy ensures comprehensive scene understanding. Extensive experiments demonstrate the superiority of our DiaRet against existing SOTA methods.

ETHICS STATEMENT

This work complies with the ICLR Code of Ethics. Our research uses only publicly available remote sensing datasets (RSICD and RSITMD), which contain satellite/aerial imagery and associated textual descriptions. These datasets do not include personally identifiable information, human subjects, or sensitive content. The images depict general geographic scenes (e.g., urban areas, harbors, campuses) and are widely used in the remote sensing community for research purposes.

THE USE OF LARGE LANGUAGE MODELS

Under the policy of ICLR, a large language model was used solely to polish the language of the manuscript, improving grammar, fluency, and clarity. It did not assist in scientific reasoning, or contribute to technical content. All ideas, methods, experiments, and conclusions are the original work of the authors. The final manuscript was thoroughly reviewed, verified, and approved by all human authors to ensure scientific accuracy and integrity.

REPRODUCIBILITY STATEMENT

We ensure reproducibility by using standard public datasets (RSICD and RSITMD) with fixed train/val/test splits as defined in prior work. All implementation details, including model architecture, loss formulations, optimizer, evaluation metrics, and hardware, are fully documented in the paper. Upon acceptance, we will release code, training scripts, and processed dialogue results.

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

# A  APPENDIX

## A.1  THE USE OF LARGE LANGUAGE MODELS

Under the policy of ICLR, a large language model was used solely to polish the language of the manuscript, improving grammar, fluency, and clarity. It did not assist in scientific reasoning, or contribute to technical content. All ideas, methods, experiments, and conclusions are the original work of the authors. The final manuscript was thoroughly reviewed, verified, and approved by all human authors to ensure scientific accuracy and integrity.

## A.2  QUALITATIVE RESULTS

We present qualitative results of top-5 retrievals for both text-to-image and image-to-text retrieval on the RSITMD dataset. As shown in Figure 8 and Figure 9, our model retrieves semantically consistent image-text pairs, demonstrating effective cross-modal alignment in remote sensing scenarios. In text-to-image retrieval, the model accurately captures the intent of diverse queries, retrieving images that reflect both global scene layouts and fine-grained spatial configurations. For image-to-text retrieval, the captions closely describe the visual content, capturing dominant land-use patterns, infrastructure types, and structural characteristics. In most cases, ground-truth results frequently appear in the top-1 and top-2 results, while lower-ranked entries maintain partial relevance (e.g., matching scene type or dominant structure), indicating robustness to semantic variations and confirming that our method effectively bridges the gap between natural language and remote sensing imagery.

| Query | Top-5 Images |
|---|---|
| On both sides of the dark blue river are white boats. |  |
| There is a white pattern on the orange desert, like a spray in the sea. |  |
| Two large ships loaded with cargo were moored on both sides of the gray port. |  |
| One side of the four ping pong balls is bare. |  |
| some planes are in an airport near several buildings and a parking lot. |  |
| On the apron surrounded by the airport runway, there is a Y-shaped terminal, and some planes are parked orderly. |  |
| A shopping district lies between two straight roads. In the middle of the shopping district is a playground. |  |
| Many buildings and green trees are in school with three playgrounds. |  |
| Two white cars and two white cars beside the house. |  |
| We can see a river, a meadow, a tree and a road around the house, as well as a swimming pool. |  |

Figure 8: Top-5 retrieved results for text-to-image on RSITMD Test Set.

| Query | Top-5 Captions |
|-------|----------------|
|  | Some buildings of a school and football fields.
Some buildings of a school and football fields.
Some buildings and football fields are in the school.
The land is a well-equipped school.
There are some buildings and football fields in the school. |
|  | The cottages with car and courtyard are on the road.
Here is a peaceful neighborhood where the house is lawn.
several houses are built by the road in this peaceful neighborhood.
the villas distribute on the either side of a road.
A cottage with cars and courtyards by the side of the road. |
|  | two bridges are across the jade green river one of which has no any cars.
Three cars were driving on the bridge opposite the green river.
Three cars drove across the green river on the bridge.
many cars are on a bridge over a river with many green plants in two sides.
A huge straight bridge spans the river. |
|  | Next to the gray conference center in the square, the parking lot was packed with cars.
Next to the square gray conference center is a parking lot full of cars.
the center is surrounded by dark green trees.
Red and green vegetation is on both sides of the gray conference center of the square.
a center building composed of three buildings is near many green trees. |
|  | here lies a well designed round center with white roof besides a row of houses.
Circular white buildings and some square buildings are surrounded by streets and houses with red roofs, as well as parked cars.
The circular building with a white center is surrounded by trees and several buildings.
a white circle center building is near two red buildings with many cars and several green trees.
the white round center is near a row of red buildings and the parking lot. |
|  | The parking factory is located in a triangle surrounded by two roads and a railway.
A factory with a parked car is located in the triangular area surrounded by two roads and railways.
The white and grey factories are located in the Triangle industrial area between the two roads.
White and grey plants are located in the industrial triangle between two roads.
the triangle industrial cintains several white and grey warehouse. |
|  | The clear pond is surrounded by yellow land and sparse trees, like a glove.
Sparse trees and national roads around the blue pond.
The lake is lying on a desert tree.
The dust Road on Damm extends across this green pond and divides it into two surrounded.
A regular pond is divided into two by a white road. |
|  | The school has a parking lot between the forest and the road, and there are buildings on the lawn, separated by paths.
this smart campus located next to a lush wood contains a parking lot and teaching buildings.
This vast area is a WELL-equipped school.
The dust Road on Damm extends across this green pond and divides it into two surrounded.
it is a peaceful campus with several teaching buildings surrounded by lush trees. |
|  | the roundabout joining six roads surround several buildings.
The streets are surrounded by green lawns and white corridors.
Seven streets are connected to circular squares near some buildings.
There is a service area with three circles at the corner of the viaduct.
There are three circles of service areas at the corner of the overpass. |
|  | There is a small tennis court and surrounded by some plants and some houses beside.
There is a small tennis court with some plants.
There is a small tennis court surrounded by plants.
four small red courts besides the big playground.
The streets surrounded by houses lie in the shade of trees. |

Figure 9: Top-5 retrieved results for image-to-text on RSITMD Test Set.