# OpenReview forum: "Interactive Remote Sensing Retrieval via Dialogue-Guided Intent Refinement and Attribute Reasoning"
_ICLR.cc/2026/Conference — ICLR 2026 Conference Withdrawn Submission_

### Official Review · Reviewer_6tpm · 2025-10-27

**Soundness:** 2
**Presentation:** 3
**Contribution:** 2
**Rating:** 4
**Confidence:** 4

**Summary:**

This paper introduces DiaRet, a dialogue-driven framework for remote sensing image-text retrieval that addresses the limitation of single-turn queries by simulating multi-round user-model interactions. The method leverages a multimodal large language model (MLLM) to iteratively refine ambiguous user intents and incorporates a fine-grained attribute reasoning module to generate structured supervision from dialogue history.

**Strengths:**

The paper convincingly challenges the unrealistic assumption in existing RSITR works that a single textual query suffices to capture complex user intent in remote sensing scenarios.

The integration of hierarchical caption reconstruction enhances both the semantic richness and discriminability of query representations.

**Weaknesses:**

1. While the paper claims to “pioneer a multi-turn interactive paradigm,” similar dialogue-based refinement strategies have already been explored in general-domain retrieval (e.g., ChatIR, PlugIR). The contribution appears to be primarily an adaptation to remote sensing rather than a fundamentally new paradigm.
2. For an interactive system, latency per turn and total session time are critical. The paper reports no inference timing, FLOPs, or computational overhead vs. static baselines.
3. The total loss in Eq. (10) uses equal weights without justification. Was this choice validated via ablation? Also, key implementation details for the IRR loss (e.g., masking ratio, vocabulary size/preprocessing) are omitted.
4. DiaRet uses powerful foundation models (Qwen-VL-Max/Qwen-Max) that likely benefit from vast external knowledge, while most baselines are trained under closed supervision.

**Questions:**

Please refer to the "Weaknesses" section.

---

### Official Review · Reviewer_WRaZ · 2025-10-27

**Soundness:** 2
**Presentation:** 3
**Contribution:** 2
**Rating:** 4
**Confidence:** 5

**Summary:**

The authors propose DiaRet, an interactive retrieval framework for remote sensing imagery that uses simulated multi-turn dialogues to disambiguate user queries. By leveraging a multimodal LLM and introducing a structured attribute reasoning module, the method aims to bridge the gap between vague initial queries and precise retrieval needs.

**Strengths:**

Clear motivation

Consistent performance improvements

**Weaknesses:**

See Questions

**Questions:**

The core idea—using dialogue to refine queries—is not new in IR literature. The novelty lies in its application to remote sensing, but the paper overstates this as a “pioneering paradigm.”

In Eqs. (1)–(2), the dynamic intent refinement uses top-S and top-K sampling, but their values and sensitivity are not reported.

Using Qwen-VL-Max gives DiaRet a significant advantage in knowledge and generation quality over conventional fine-tuned models. How do you verify that the gains come from the proposed framework and not just a more powerful LLM?

The paper does not provide results on a hard-negative test subset. The reported improvements in R@K might arise primarily from better matching of easy positives rather than demonstrating superior discrimination among highly similar negatives that challenge global feature-based methods.

---

### Official Review · Reviewer_jpT4 · 2025-10-28

**Soundness:** 3
**Presentation:** 3
**Contribution:** 3
**Rating:** 6
**Confidence:** 5

**Summary:**

This paper addresses the issues of semantic ambiguity and unclear user intent in existing research on remote sensing image-text retrieval (RSITR), which typically relies on raw text descriptions and fails to account for the inherent inter-class similarity and multi-scale features of remote sensing imagery. To this end, a novel dialogue-driven framework, DiaRet, is proposed to pioneer a multi-turn interactive paradigm for RSITR by leveraging conversational interactions to clarify ambiguous user retrieval intents and enhance retrieval accuracy through fine-grained attribute reasoning. The framework employs Multimodal Large Language Models (MLLMs) to enable Dynamic Intent Refinement (DIR), progressively clarifying user intent via context-aware question-answering interactions. Additionally, an LLM-based Fine-grained Attribute Reasoning (FAR) module is introduced to transform dialogue history into structured editing instructions and key visual keywords, thereby optimizing the model’s focus on detailed visual attributes. Extensive experiments on the RSICD and RSITMD benchmarks demonstrate that DiaRet achieves state-of-the-art performance, validating its superiority and practicality in real-world applications.

**Strengths:**

This paper innovatively proposes a novel dialogue-driven framework (DiaRet) for the remote sensing image-text retrieval task, introducing multi-turn interaction to refine user intent and incorporating a fine-grained attribute reasoning module specifically designed to address the unique challenges of remote sensing imagery. The effectiveness of the approach is validated through comprehensive comparative and ablation studies, demonstrating improved accuracy and practicality in remote sensing image retrieval. Furthermore, the work provides new insights into the development of interactive retrieval systems.

**Weaknesses:**

Although the proposed method is novel in design and demonstrates superior performance, several aspects remain open to improvement. While the multi-turn dialogue mechanism enhances retrieval accuracy, it introduces additional computational overhead and response latency. The authors have not provided a quantitative analysis of inference time or resource consumption, nor included an efficiency comparison with single-turn retrieval, which may limit its applicability in real-time scenarios.

**Questions:**

1.	Does the dialogue-based retrieval approach proposed in the article introduce additional computational complexity and time overhead?
2.	During the dialogue interaction process, does the phrasing of the questions have an impact on the retrieval results?
3.	The paper mentions generating scene descriptions at three levels, but does not clarify whether questions are asked about the overall structure first or about fine-grained details, nor does it discuss whether the order of questioning may influence the final retrieval results.
4.	The paper uses an MLLM to simulate user responses, which are assumed to be accurate. However, in real-world scenarios, users may provide ambiguous or incomplete answers—could this affect the retrieval performance?
5.	The DIR module already demonstrates strong perception and reasoning capabilities. Have the authors considered extending it into a more autonomous AI agent?

---

### Official Review · Reviewer_Gw6R · 2025-10-31

**Soundness:** 2
**Presentation:** 2
**Contribution:** 2
**Rating:** 2
**Confidence:** 3

**Summary:**

This paper proposes DiaRet, an interactive remote sensing image-text retrieval (RSITR) framework that introduces a dialogue-guided paradigm for refining user intent. The method consists of two main modules: Dynamic Intent Refinement (DIR), which engages in multi-turn question-answering to progressively clarify vague textual queries, and Fine-grained Attribute Reasoning (FAR), which extracts keywords and generates structured “atomic instructions” for targeted optimization. Experiments on RSICD and RSITMD datasets show modest improvements over several existing approaches such as RemoteCLIP and PlugIR.

While the paper explores an interesting direction by introducing dialogue-based refinement into RS retrieval, the motivation, methodological clarity, and experimental significance are not sufficiently convincing. The practical value of RS image-text retrieval itself is questionable, and the proposed modules (DIR, FAR) are not well justified or clearly connected.

**Strengths:**

1) The idea of applying an interactive dialogue-driven paradigm to remote sensing retrieval is conceptually interesting.
2) The multi-scale caption reconstruction aligns with the multi-scale nature of RS imagery.
3) The writing style is generally clear, and the experimental setup follows standard RSITR evaluation protocols.

**Weaknesses:**

1) Limited real-world value: Both text-to-image (T2I) and image-to-text (I2T) retrieval have limited applicability in remote sensing, especially I2T, which lacks realistic use cases. Text descriptions in RS caption datasets are typically written after observing the images, so performing retrieval based on such textual queries (as if from memory) is not practically meaningful.
2) Unclear module relationship: The relationship and data flow between DIR and FAR are poorly explained. Fig. 2 is overloaded and difficult to interpret.
3) Lack of motivation for key components: The reason for keyword extraction and masked learning (IRR) is not well justified. The ID loss lacks relevance for the RS domain.
4) Ambiguous task formulation: The method seems designed for text-to-image retrieval, yet results for image-to-text retrieval are reported without describing how it works.
5) Inaccurate reporting: Table 1 highlights a wrong best result (RSICD T2I R@1).
6) Minor performance gains: Improvements over PlugIR or RemoteCLIP are marginal.
7) Unfair or uninformative ablations: 1) The “0–1 round” comparison only proves that vague queries perform worse, not that dialogue refinement is inherently superior. 2) The “multi-scale feature extraction” ablation is trivial and adds little insight.
8) Missing visual evidence: Figures 6 & 7 show only final retrievals, not the dialogue refinement process—thus failing to visualize the claimed advantage.

**Questions:**

1) Please clarify the actual use cases of remote sensing image-text retrieval.
2.	How exactly do DIR and FAR interact? Is FAR applied after all dialogue rounds or at each iteration?
3.	What is the concrete motivation behind keyword masking—is it inspired by masked language modeling or intended to improve fine-grained alignment?
4.	The paper claims that the ID loss enforces intra-modal consistency, but it is unclear why such intra-modal clustering is needed for a retrieval task focused on cross-modal alignment, and whether removing the ID loss would actually degrade the results.
5.	How is image-to-text retrieval implemented?
6.	Could you provide visualizations of the dialogue process (intermediate rounds, query updates, and retrieval rank improvements)?
7.	Please verify and correct Table 1 results (especially RSICD T2I R@1).
8.	Have you considered comparing with a baseline that uses a single finely written query instead of iterative refinement to ensure fairness?

---

### Note · Authors · 2025-11-14

I have read and agree with the venue's withdrawal policy on behalf of myself and my co-authors.